# Single-Cell Profiling Reveals Global Immune Responses During the Progression of Murine Epidermal Neoplasms

**DOI:** 10.3390/cancers17081379

**Published:** 2025-04-21

**Authors:** Xiying Fan, Tonya M. Brunetti, Kelsey Jackson, Dennis R. Roop

**Affiliations:** 1Department of Dermatology, University of Colorado Anschutz Medical Campus, 12700 E. 19th Ave., Room 4007, Aurora, CO 80045, USA; kelsey.jackson@cuanschutz.edu; 2Gates Institute, University of Colorado Anschutz Medical Campus, Aurora, CO 80045, USA; 3Department of Immunology and Microbiology, University of Colorado Anschutz Medical Campus, Aurora, CO 80045, USA; tonya.brunetti@cuanschutz.edu

**Keywords:** scRNA-seq, tumor microenvironment, immune response

## Abstract

The immune system plays a crucial role in either fighting or supporting tumor growth. In this study, we analyzed immune cells in control skins and skin tumors using single-cell RNA sequencing. We identified 15 different immune cell types and observed significant changes in their composition during skin tumor progression. Macrophages became the dominant immune cells in tumors and acted as key regulators of immune interactions, while other immune cells, including Langerhans cells and DETC, decreased. We also found that tumors activated specific signaling pathways, including Jak2/Stat3, and increased immune-suppressing cells like Tregs and M2 macrophages. These changes help tumors evade the immune system. Our findings provide new insights into how the immune system responds to skin tumor progression and may help guide future treatment strategies.

## 1. Introduction

Epithelial cancers constitute approximately 90% of all human malignancies. Among these, cutaneous squamous cell carcinoma (cSCC), arising from malignant proliferation of keratinocytes of the skin [1], is the second most common cancer in the United States with an annual incidence of over one million [2]. Unfortunately, the incidence of cSCC has continued to dramatically rise over the past three decades. Although excision can be curative, up to 4% of cSCC patients develop nodal metastasis and approximately 1.5% die from the disease [3]. The recent approval of cemiplimab, an inhibitor of programmed death 1 (PD-1) for advanced cSCC is promising [4], but not all patients respond to this immune checkpoint inhibitor, and those who respond may develop resistance over time [5,6]. The tumor microenvironment (TME) plays a critical role in anti-tumor immunity [7,8], which may result in immunotherapy being ineffective. Therefore, there is an urgent need to better understand the mechanisms underlying primary and acquired resistance to immunotherapy and better characterize immune cells in the tumor microenvironment during cSCC progression.

Immune cells in the TME can attack and destroy tumor cells, but can also be co-opted by the tumor to promote growth and metastasis [9]. Given the dual roles of various immune cells, tumor-suppressing and tumor-promoting [10], the complexity of the immune landscape in the TME is a significant area of research. The concept of an immune checkpoint blockade that has emerged from this research revolutionizes the cancer treatment field [11]. However, the effectiveness of such immunotherapies is variable and immune cells can contribute to immunotherapy resistance. Recent studies suggest that modulating immune cells through pharmacological intervention offers a promising strategy for cancer treatment [12]. Thus, critical insights from current research highlight a need for a thorough understanding of immune cell function within the TME.

To this end, we applied single-cell RNA sequencing (scRNA-seq) on CD45^+^ cells within the TME to characterize immune cell responses during cSCC progression, since scRNA-seq is a key method to comprehensively characterize the intra-tumoral heterogeneity within many cancer types [13,14,15]. Given the essential role of preclinical mouse models in cancer research [16,17], we inoculated tumor cells into mice and isolated CD45^+^ immune cells from control skins and skin tumors. Single-cell sequencing was then performed to characterize the overall immune cell responses to skin tumorigenesis during cSCC progression. We identified 15 CD45^+^ immune cell clusters, which broadly represent the most functionally characterized immune cells in murine skin.

Globally, we observed a remarkable increase of interferon response and JAK/STAT3 signaling in skin tumors. Of note, skin tumor progression reprogramed immune cells, and led to a marked increase in the relative percentages of macrophages, cDC2, mDC, Tregs, and Neu. Macrophages were the largest group of immune cells found in skin tumors. Particularly, M2-macrophages, which have been shown to promote tumor growth, were the main macrophage type in skin tumors. More importantly, macrophages emerged as the predominant communication ‘hub’ in skin tumors, highlighting the importance of macrophages during skin tumor progression. In contrast, other immune cell clusters decreased during skin tumor progression, including DETC, γδT, ILC2, and LC. In addition, skin tumor progression dramatically upregulated JAK/STAT3 expression and the interferon response across various cell types. Meanwhile, a pronounced infiltration of Tregs in skin tumors created an immunosuppressive microenvironment, consistent with the elevated expression of the Stat3 pathway in skin tumors.

Our scRNA-seq reveals the immune cell landscape during cSCC progression, and enhances the understanding of the molecular effects of tumorigenesis on the cutaneous immune system. An in-depth understanding of these molecular signatures during tumor progression makes immunotherapy an effective and promising tool in cancer treatment. In summary, our study provides a blueprint of the molecular signatures of immune cells in the tumor microenvironment and identifies potential immunotherapies for their clinical utilization.

## 2. Methods

### 2.1. Mice

All mice were housed under specific pathogen-free conditions in the vivarium facility of the University of Colorado Anschutz Medical Campus. Protocols and procedures were approved by the Institutional Animal Care and Use Committee at the University of Colorado Anschutz Medical Campus. All animal experimentations were approved by the IACUC of the University of Colorado [protocol 00073].

### 2.2. Cell Lines and In Vivo Cell Injections

The 2323 cell line was derived from our mouse model K14rtTA/tetOCre/Kras^G12D^/P53^−/−^ in the escape phase as described previously [18]. 2323 cells were cultured in a DMEM media containing 10% fetal bovine serum and 1% penicillin/streptomycin. For tumor cell inoculation, female and male C57BL/6 mice (Jackson Laboratories) between 8 and 10 weeks old were injected with 0.2 M 2323 cells suspended in 200 μL PBS on their flank. Mice were euthanized when tumors reached a size of 100–200 mm^3^.

### 2.3. Control Skin and Skin Tumor Digestion and Flow Cytometry

Control skin and skin tumors were excised from mice, minced, and dissociated with a 0.5% collagenase IV (Worthington, Lakewood, CA, USA, LS004188) at 37 °C for 2 h. Cells were spun down for 5 min at 300 g and strained through a 70 μm strainer before counting and downstream analysis. Next, cells were stained with CD45-APC (Life technologies, Carlsbad, CA, USA, MCD4505) for 30 min on ice and then washed with PBS. Live/dead exclusion was performed using 0.1 mg/mL DAPI (4′,6-diamidino-2′phenylindole dihydrochloride). Then, single cell suspension was used to sort for CD45^+^/DAPI^−^ immune cells on a Beckman Coulter XDP100 cell sorter at the University of Colorado Cancer Center Flow Cytometry Shared Resource.

### 2.4. Single-Cell RNA-Seq

Single-cell RNA-seq was performed according to the manufacturer’s instructions (10x Genomics) at Genomics and the microarray sequencing core. The sorted CD45^+^ cells from two 2323 tumors and two C57 control skins were loaded into a Chromium Single Cell 3′ Solution Capture System (10x Genomics). The processed libraries were sequenced on Illumina NovaSeq X at the Genomics Shared Resource at the University of Colorado Anschutz Medical Campus. The sequencing reads were mapped to mouse genome (mm10) using 10x Genomics Cell Ranger (version 7.1.0) [19]. Different samples were aggregated using the “cellranger aggr” command with normalization using mapped reads.

Data were further processed and filtered using Seurat (version 5.0.1) in R (version 4.3.2) to perform quality control filtering, normalization, and clustering [20]. Cells were removed if there were fewer than 100 genes detected, a UMI count less than 1000, a percentage of mitochondrial genes more than 15%, or log10GenesPerUMI (the gene numbers per UMI) less than 0.8. Genes were excluded if they were detected in fewer than 3 cells. Following filtering, the UMI counts were normalized to library size. Harmony (version 1.1.0) was used to integrate two control skins (C57_1 and C57_2) and two skin tumor samples (2323_1 and 2323_2), resulting in a batch-corrected expression matrix for all cells. The new integrated matrix was used for scaling and the principal component analysis (PCA). Now-linear dimensional reduction was performed using the uniform manifold approximation and projection (UMAP) method. Next, scDblFinder (version 1.16.0) was used to remove doublets. After filtering, a total of 11,525 cells, with 5496 cells from C57 control skins and 6029 cells from 2323 skin tumors, were used for the downstream analysis.

#### 2.4.1. Cell Clustering Analysis

To identify cell clusters, PCA was first performed on 2000 highly variable genes. Significant PCs were identified using an Elbowplot and the first 25 PCs were used for clustering with the Louvain modularity-based community detection algorithm to generate cell clusters (FindClusters function, 15 immune cell clusters with resolution = 0.3).

#### 2.4.2. Identification of Differentially Expressed Genes and Marker Genes

The differentially expressed genes (DEGs) in specific cell types were identified by using FindAllMarkers functions in Seurat to identify differentially expressed genes between clusters. After identifying transcripts enriched in each of our 15 initial clusters (p.adj < 0.05), we manually assigned class identity based on comparison to well-established marker genes

#### 2.4.3. Functional Enrichment Analysis

We used the package clusterProfiler enricher function to perform Gene Ontology (GO) enrichment analysis. Genes with |log2FC| > 2 and p.adj < 0.05 were used as input for the enrichment analysis. The KEGG pathway and hallmark pathway from the molecular signature database (MsigDB) were extracted for gene set variation analysis to speculate the enrichment scores for the samples.

#### 2.4.4. Cell Communication and Signaling Pathways

CellChat is a tool that can quantitatively analyze intercellular communication networks from scRNA-seq data (http://www.cellchat.org). The interactions were identified and quantified based on the differentially expressed ligands and receptors for each cell group. The analysis was conducted using the CellChat R package (version 1.5.1) [21]. The CellChatDB.mouse receptor interaction database was used in the analysis. The functions of identifyOverExpressedGenes and identifyOverExpressedInteractions in CellChat were utilized to discover over-expressed genes and their accompanying interactions. The communication probability was determined by using the computeCommunProb, filterCoomunication, and computeCommunProbPathway functions. The netAnalysis_contribution function was used to calculate the contribution of each ligand–receptor pair to the whole signaling pathway. The extractEnrichedLR function was used to retrieve significant ligand–receptor pairs and their signaling genes for a certain signaling pathway. The netVisual_bubble tool was utilized to graphically plot the significant ligand–receptor interactions.

#### 2.4.5. Immunofluorescent Staining

Sections of OCT-embedded tumor samples were cut 10 µm using a cryostat. Images were analyzed using Nikon NIS analysis software (NIS-Elements AR 4.5). OCT sections were fixed for 10 min in 4% PFA in PBS and washed three times for 5 min in PBS. The following block solution was used: 2.5% NGS, 2.5% NDS, 2% gelatin and 0.3% Triton X-100 in PBS. The following primary antibodies were used: CCL8 (ThermoFisher, Waltham, MA, USA, #PA5-87000), F4-80 (Abcam, Cambridge, UK, #ab6640), and CXCL10 (ThermoFisher, #701225), Ifng (ThermoFisher, Waltham, MA, USA, #14-7311-81), Cd4 (Biolegend, San Diego, CA, USA, #76870). Sections were incubated in primary antibodies overnight at 4 °C. Finally, the sections were incubated with fluorophore-conjugated secondary antibodies. The images were acquired with a Nikon microscope.

#### 2.4.6. Statistics and Reproducibility

Two replicates of 2323 tumor and C57 control were used for scRNA-seq. Statistical analyses were performed using Seurat (version 5.0.1) in R (version 4.3.2). Data normalization was conducted using Seurat’s log-normalization method. For dimensionality reduction, PCA was applied, and significant components were selected using JackStraw analysis. Clustering was performed using the Louvain/Leiden algorithm. Deferential expression analysis was carried out using Seurat’s default Wilconxon rank-sum test, with *p*-values adjusted for multiple testing using the Benjamini-Hochberg method. Differences in immune cell populations between control and tumor tumors were assessed using the chi-square test. Statistical significance was defined at *p* < 0.05.

Data and code availability: All data were deposited in the NCBI’s Gene Expression Omnibus database (GSE280070).

## 3. Results

### 3.1. Unbiased Clustering of Single-Cell Transcriptomes Confirms Known Cutaneous Immune Cell Populations

#### 3.1.1. Identification of 15 Distinct Immune Cell Populations

To explore and compare immune cell responses during skin tumor progression, we inoculated tumor cell line 2323, established from a ktiKPP tumor graft, subcutaneously into C57BL/6 mice. They were euthanized two weeks after the inoculation when skin tumors had grown. Then, we collected control skins from uninoculated C57 animals and skin tumor samples from inoculated animals, and isolated single cells enzymatically from collected samples. Next, we flow sorted CD45^+^ immune cells and performed scRNA-seq using droplet-based microfluidics (10x Genomics). An outline of these procedures is shown in Appendix A. The total number of single cells initially isolated from each group was Ctrl_1: 3499; Ctrl_2: 3862; Tumor_1: 5176; and Tumor_2: 2258 (Appendix A).

Using Seurat [22], data was first processed through quality control filtering, normalization, and clustering. Next, Harmony [23] integration, dimensional reduction, principal component analysis, and unsupervised clustering were performed on a total of 11,525 cells to yield the initial 15 clusters at resolution 0.2 (Appendix A). To define these clusters, we utilized FindAllMarkers functions in Seurat to identify differentially expressed genes between clusters. After identifying transcripts enriched in each of our 15 initial clusters (p.adj < 0.05), we manually assigned class identity based on comparison to well-established marker genes (Appendix A). One cluster (cluster 10) expressed high levels of fibroblast marker genes, including Col3a1, Col6a1, and Col6a2, indicating an apparent fibroblast contamination (Appendix A). Another small cluster (cluster 14), having only 20 cells, didn’t express a clear immune cell class. Therefore, we excluded these two small clusters in our later analysis.

After the exclusion, we processed data through unsupervised clustering again and produced another 15 immune cell clusters at a higher resolution 0.3, each showing robust representation of immune cell types across both control skins and skin tumors (Figure 1A,B). The heatmap of the top 20 differentially expressed genes for each cluster showed distinct transcriptomic profiles for each cluster and allowed cell cluster identification (Figure 1C). Of these 15 immune cell clusters, there were two proliferative clusters, five antigen-presenting cell (APC) clusters, four T cell clusters, and four other non-APC clusters.

##### Two Proliferative Immune Cell Clusters

Two immune cell clusters highly expressed proliferative markers Top2a and Ki67 (Figure 1D and Appendix A). Thus, we defined these two proliferative clusters as Prolif.1 and Prolf.2. To better define these two proliferative clusters, we found that MHC II genes such as H2-Ab1, H2-Ab, and H2-Eb1 were highly expressed in Prolif.1, suggestive of one proliferative antigen-presenting cell (APC) cluster (Appendix A). Further analysis of Adgre1 and Mrc1 expression showed these two genes expressed in Prolif.1, suggesting that this proliferative APC cluster was a proliferating macrophage cluster (Appendix A). In addition, Prolif.2 expressed T cell marker genes such as Cd3g, Cd3e, Trac, Cd4, Cd8a, and Gzmb, suggestive of a proliferating T cell cluster (Appendix A). However, the role of these two proliferating clusters during cSCC progression needs further investigation.

##### Five APC Clusters

Five immune cell clusters had a high expression of MHC II transcripts, such as H2-Ab1, H2-Aa, and H2-Eb1, suggesting they were APC clusters. One of these five immune cell clusters appeared consistent with macrophages (“Mac”), based on its expression of Adgre1, Itgam, Fcgr1, and Cd68 [24]. This cell cluster also expressed tissue-remodeling macrophage markers such as Mrc1 (Figure 1D,E). A second cell cluster with elevated Cd207, Cd24a, and Epcam expression was identified as Langerhans cells (referred to hereafter as “LC”; Figure 1D,E).

The remaining three APC clusters displayed increased expression of H2-Ab1, H2-Eb1, and Flt3, lacked the macrophage marker Folr2, and were consistent with dendritic cells. To better understand these three clusters, we identified differentially expressed genes for each, in comparison to the macrophage and Langerhans cells. One cluster showed elevated Xcr1, Irf8, and Clec9a, and an absence of Irf4, Itgam, and Sirpa (Figure 1D,E, Appendix A), identifying them as likely conventional type 1 dendritic cells (referred to hereafter as “cDC1”) [25]. The second cluster showed the upregulated expression of Mgl2, Sirpa, and Irf4, and an absence of cDC1 markers (Irf8, Batf3, and Cd207) consistent with conventional type 2 dendritic cells (cDC2s) (Figure 1D,E, Appendix A). The third population was characterized by elevated migration and activation markers Fscn1, Cacnb3, Ccr7, Cd40, Tmem123, and Cd274 (Figure 1D,E, Appendix A), suggestive of a group of migratory/mature dendritic cells (referred to hereafter “mDC”) [26,27,28].

##### Four Non-APC Clusters (T-Cell Clusters)

Of the remaining eight non-APC CD45^+^ cell clusters, four expressed elevated levels of Cd3e/g suggestive of T cells. The first of these three clusters was characterized by a high expression of Cd3e/g, Trdc, Tcrg-C1, Thy1, Nkg7, Fcer1g, and Il2rb, and a lack of Cd4/Cd8a expression (Figure 1D,E); this is consistent with dendritic epidermal T cells (“DETC”), a population of embryonically derived, tissue-resident γδT cells that function in cutaneous immune surveillance [29,30]. We identified another cluster as dermal γδ T cells (referred to hereafter “γδT”) based on strong Rora expression and intermediate expression of Cd3e/g, Trdc, and Tcrg-C1/2/4, and a lack of Cd4 and Cd8a expression (Figure 1D,E) [31]. The third cluster presented a broad expression of Cd3d, Trac, Trbc1, Cd4, and Cd8a, suggestive of T cells (referred to as “T”, Figure 1D,E). The fourth cluster with high expression of Foxp3 was defined as regulatory T cells (referred to “Tregs”, Figure 1D,E).

##### Four Other Non-APC Clusters

We identified another cluster as type 2 innate lymphoid cells (ILC2) based on high expression of Gata3, Rora, Il7r, and Il5, coupled with very low levels of Cd3d/e/g, Eomes, Rorc, and Cd4 (Figure 1D,E). This group was also negative for natural killer (NK) and NK T cell markers such as Klra7, Klra8, and Klrb1. An NK cluster with elevated expression of NK-related markers such as Gzma, Klra8, Klra7, Klrb1c, Eomes, and Nrc1 was also identified (Figure 1D,E).

Finally, a cluster of mast cells (referred to as “Mast”) was defined by Gata2, Ms4a2, Kit, Mcpt8, Mcpt4, and Itgam expression, and neutrophils (referred to as “Neu”) were defined by Cd14, S100a8, S1009, Csf3r, Cebpd, Slc11a, and Spi1 expression (Figure 1D,E).

In summary, our scRNA-seq data revealed 15 key immune cell clusters in control skins and skin tumors. We utilized these immune cell clusters to gain a better understanding of how these immune cells respond to skin tumor progression. Our study will shed light on improving immunotherapy against tumor progression.

### 3.2. Tumor Progression-Induced APC-Dominant Shifts into Mouse Skin

Although control skin and skin tumor datasets contained the same basic cell types, their immune cell composition showed substantial differences. To evaluate the relative abundance of 15 immune cell types during skin tumor progression, we examined the percentage of each cell cluster in both control skin and skin tumor datasets. Our data showed that skin tumor progression led to a marked increase in relative percentages of macrophage, Treg, cDC2, Neu, mDC, and two proliferating cell populations (Prolif.1 and Prolif.2) (Figure 2A,B). Meanwhile, immune cell clusters decreased during skin tumor progression, including DETC, γδT, ILC2, and LC (Figure 2A,B). Thus, the dramatic changes in immune cell types induced APC-dominant shifts during skin tumor progression.

### 3.3. Skin Tumor Progression Reprogramed CD45^+^ Cell Transcriptomes Toward JAK/STAT Signaling or Interferon Responses

In addition to immune cell composition changes during skin tumor progression, we also compared the transcriptional differences across the immune cell clusters in control skins and skin tumors. We compared transcripts differentially expressed (|log2FC| > 1 and p.adj < 0.05) in control skins and skin tumors to identify genes significantly changed during skin tumor progression. Gene set variation analysis showed that enriched gene signatures in skin tumors included the interferon-gamma response, the interferon-alpha response, and JAK/STAT3 signaling (Figure 2C). Further, IFN-γ and IFN-a signatures dramatically increased across immune cell clusters in skin tumors (Figure 2D,E). In addition, upregulation of JAK/STAT3 signaling was found globally across the immune cell types in skin tumors (Figure 2F).

Although the immune cell composition and the overall transcriptomics changed during skin tumor progression, the specific immune cells responding to skin tumorigenesis are separately described below.

### 3.4. Resident and Recruited Macrophages Dramatically Increased in Skin Tumors

Macrophages have emerged as a critical regulatory cell type in the tumor microenvironment, facilitating tumor initiation, progression, and metastasis [32]. As the most abundant immune-related stromal cells, macrophages provide a favorable milieu for tumor cells [33]. Similarly, our scRNA-seq data revealed that macrophages were one of the most abundant immune cell types in skin tumors. Thus, a better understanding of the roles of tumor-infiltrating macrophages during skin tumor progression is needed. Further, identifying the specialized subpopulations of macrophages in response to skin tumor progression may represent important therapeutic targets.

In our study, we identified one distinct cluster significantly expressing MHCII molecules and macrophage markers (Appendix A). As shown in Figure 2B, macrophages dramatically increased in skin tumors compared to control skins. In addition, we utilized the CellChat tool, analyzing our high-resolution scRNA-seq data to explore the cell–cell communication specific to cSCC. Our analysis showed a significant change in the number of interactions between the control and tumor groups (Figure 3A). Of note, macrophage populations emerged as the predominant communication ‘hub’ in skin tumors (Figure 3A and Appendix A). In addition, CCL and CXCL signaling pathways were among the top prominent outgoing and incoming signaling patterns in skin tumors (Appendix A). An in-depth exploration of the CCL and CXCL signaling pathways indicated that each ligand–receptor interaction from macrophages to other cell types contributes to signaling. Notable interactions include the Pf4/Cxcl9/Cxcl4/Cxcl10-Cxcr3 pairs, Cxcl2-Cxcr2, Cxcl16-Ccr6, Ccl9/Ccl8/Ccl7/Ccl5/Ccl3-Ccr1, and Ccl8/Ccl7/Ccl6/Ccl2-Ccr2 (Figure 3B). These results highlighted the importance of macrophages in skin tumor progression.

To better understand the potential functions of macrophages in the TME of skin tumors, we further divided this cluster into three sub-clusters: sub-cluster 0, 1, and 2. Although the absolute cell numbers of macrophages increased in skin tumors, we also investigated the change in proportion of each sub-cluster in skin tumors. Sub-cluster 2 dramatically increased in skin tumors, but not sub-cluster 0 and 1 (Figure 3C), suggesting that sub-cluster 2 was the main cell sub-cluster responding to skin tumor progression.

To further analyze and understand this sub-cluster 2 in responding to skin tumor progression, we first examined the molecular features of each sub-cluster. Sub-cluster 0 highly expressed monocyte marker genes Plac8 and Ly6c2, suggestive of monocyte-derived macrophages (Figure 3D). Sub-clusters 1 and 2 had a high M2-like score and highly expressed M2-like marker Mrc1 (Figure 3E), suggestive of M2-like macrophages. M2-like macrophages were reported to release immunosuppressive chemicals while promoting tumor cell growth, drug resistance, angiogenesis, and tissue healing [34]. In addition, M2-like macrophages were consistent with elevated STAT3 signaling in skin tumors [35].

Consistent with M2-like macrophages in skin tumors, sub-cluster 2 showed high expression of M2-like macrophage markers, including Mgl2, Cx3cr1, Vcam1, Maf, Ccl8, and Cxcl10. In contrast, sub-cluster 0 highly expressed Plac8 and Ly6c2 suggestive of monocytes-derived macrophages, while sub-cluster 1 expressed lower levels of Cx3cr1, Vcam1, and Ccl8 (Figure 3E). The Mgl2 gene, a member of the macrophage galactose-type C-type lectin family, is expressed in M2 macrophages [36]. The Cx3cr1 gene, the chemoattractant cytokine Cx3cl1 receptor, regulates the infiltration and polarization of TAMs in tumors [37]. Previous studies showed that Cx3cr1 expression contributed to macrophage survival in tumor metastasis and was correlated with poor prognosis in human cancers [38]. Vcam1, vascular cell adhesion molecule 1, promotes tumor cell invasion and metastasis in colorectal cancer [39]. Maf (c-Maf), a member of the basic leucine zipper transcription factors, controls many M2-related genes and promotes M2-like macrophage-mediated T cell suppression and tumor progression [40]. Ccl8, chemokine ligand 8, secreted by M2-like macrophages, promotes tumor growth and invasion in different cancer types [41,42]. In addition, there is increasing evidence that Cxcl10 plays a tumorigenic role, causing tumor progression and metastasis in different cancers [43,44,45]. Consistent with these reports, we observed a higher expression of Ccl8 and Cxcl10 in 2323 skin tumors by immunofluorescent staining compared to control skins (Figure 3F). Consistent with these studies, the higher expression of M2-like macrophage marker genes in our data suggests that this M2-like sub-cluster 2 contributes to skin tumor progression. Of note, high expression of Ccl8 and Cxcl10 was observed in sub-cluster 1 and 2; targeting CCL8 and Cxcl10 might provide a novel target for cSCC treatment. 

### 3.5. Activated T Cells in Response to Skin Tumor Progression

Well-established cytotoxic T cells are important effectors of anti-tumor immunity in human cancers. We identified a distinct T-cell cluster expressing Cd3d/e/g, Cd4, Cd8a, and Thy1 (Figure 1D). We did not observe a significant infiltration of this pan T cell cluster in skin tumors as shown above in Figure 2B. However, we observed inflammatory responses in skin tumors (Figure 4A). Consistently, we observed inflammatory genes, including Ifng, Stat1, Ifit2, Ifit3, Isg15, and Bst2, being highly expressed in the T cell cluster in skin tumors (Figure 4B). In addition, this T-cell cluster exhibited over-expression of exhaustion marker genes such as Tox, Pdcd1, Tigit, Havcr2, Lag3, and Ctla4 [46] in the T cell cluster in skin tumors (Figure 4B), suggesting the emergence of T-cell exhaustion in skin tumors. This is consistent with the known fact that most T cells in tumor microenvironment are exhausted [47]. Additionally, an elevated signaling of Il2/Stat5 in skin tumors (Figure 4A) is consistent with a recent report stating that Il2 regulates CD8 T cell exhaustion in the tumor microenvironment [48].

To better understand the role of this T cell cluster during skin tumor progression, we further subsetted this cluster into three sub-clusters: sub-cluster 0, 1, and 2 (Figure 4C). Of note, sub-cluster 2 was the only subgroup to respond to skin tumor progression, which dramatically increased from 4.1% in Ctrl to 20.7% in Tumor (Figure 4D). To better understand the features of this sub-cluster 2 and its functions in skin tumors, we first examined the molecular features of this sub-cluster and compared them to the other sub-clusters. We found that sub-cluster 0 had a very low expression of Cd4; sub-cluster 1 highly expressed Cd8a and Cd8b1, suggestive of Cd8^+^ T cells; sub-cluster 2 highly expressed Cd4, suggestive of Cd4^+^ T cells (Figure 4E). Further, this Cd4^+^ T cell sub-cluster 2 also highly expressed Cd70, Cd80, Cd81, Cd83, and Pdcd1 (Figure 4E). These co-stimulatory molecules highly expressed in this Cd4^+^ sub-cluster were consistent with previous research [49,50,51,52]. Additionally, we observed high expression of Pdcd1 in skin tumors (Figure 4E), highlighting the potential role of CD4^+^ T cells in checkpoint inhibitor immunotherapy. More interestingly, we observed that this CD4^+^ T cell population expressed cytotoxic marker genes such as Ifng, Nkg7, Lamp1, Lamp2, Klrc1, and Runx3 highly expressed in sub-cluster 2 and decreased expression of CD27 [53,54], suggestive of the cytotoxic functions of CD4^+^ T cells during skin tumor progression. Consistently, we observed the co-localized expression of Cd4 and Ifng in skin tumors, but not in control skins (Figure 4F). These findings highlighted the critical roles of CD4^+^ T cells in responding to skin tumor progression, which is consistent with the recognition of CD4^+^ T cells as anti-tumor effector cells in their own right [55]. Future studies on this cytotoxic subset of CD4^+^ T cells during skin tumor progression may offer insights into novel therapeutic strategies and improve clinical outcomes.

### 3.6. Activated NK Cells in Skin Tumors

Natural killer cells play an important role in host immunity against pathogens and cancers [56,57]. We identified one NK cell cluster based on its expression of Klr genes which encode molecules expressed on NK cells (Figure 5A). This correlated significantly with natural killer cell mediated cytotoxicity by KEGG pathway analysis (Figure 5B). As shown above in Figure 2B, we did not observe a significant change of NK cell composition in control skins and skin tumors. Further, we found activated NK cells in skin tumors indicated by higher levels of NK activation genes including Gzmb, Gzma, Eomes, Itga2 (Cd49b), and Klra8 (Figure 5C). Conversely, NK inactivation genes including Itga1 (Cd49a), Inpp4b, Tnfsf10, Il7r, and Cxcr6 dramatically decreased in skin tumors (Figure 5C).

To further investigate the unique features of this activated NK cell cluster in skin tumors, we examined the upregulated Klr molecules in skin tumors and identified several Klr genes including Klra8, Klre1, Klra4, Klrl2, and Klrb1c, which dramatically increased in skin tumors (Figure 5C,D). Further, we observed higher expression of interferon-activated genes including Ifi213, Ifi206, and Ifi209, which specifically expressed in skin tumors (Figure 5E), suggestive of an important role of interferon signaling in activating NK cells in skin tumors. Collectively, our data revealed an activated NK cell cluster responding to skin tumor progression and identified the unique features of this cell cluster with specific Klr gene expression.

### 3.7. Tregs

Regulatory T cells (Tregs) have been known to suppress antitumor immunity [58]. We identified one Treg cell cluster that expressed key Treg marker genes such as Foxp3, Cd4, Tnfrsf18, and Ctla4 (Figure 6A). This Treg cell cluster dramatically increased in relative abundance between control skins and skin tumors (Figure 2B), suggestive of the significant accumulation of Treg cells in skin tumors. In addition, we observed a higher expression of PD-1 (encoded by Pdcd1), Itga4, Sell, and Ccl5 in skin tumors (Figure 6B). PD-1, a well-established T cell exhaustion marker, was reported to promote the immunosuppression function of Tregs [59]. Consistent with its enhanced immunosuppressive role, Tregs in skin tumors presented higher levels of Itga4, Sell, and Ccl5. Itga4, integrin subunit alpha 4, was reported to upregulate in three T cell subsets such as Tregs in age-related tumor growth [60]. Sell, also known as CD62L, was one of the three key molecules associated with high levels of suppression mediated by human Treg [61]. Ccl5 was reported to contribute to tumor growth in a variety of cancer types [62]. For example, Ccl5 was able to recruit Treg cells in pancreatic ductal adenocarcinoma, and this recruitment was impaired by the neutralization of Ccl5 [63]. In summary, we observed a dramatic accumulation of Tregs and an increased immunosuppressive function of Tregs in skin tumors.

### 3.8. Neutrophils

Neutrophils, the most abundant myeloid cells in human blood, act as the body’s first line of defense against infection and cancers, and emerged as important regulators of cancer [64]. Growing evidence has recognized the critical role of Neutrophils in promoting tumor growth and patients with neutrophils predict poor overall survival in many types of cancers [65]. We detected one neutrophil cluster that highly expressed neutrophil markers (Figure 6C). Of note, this neutrophil cluster also largely expanded in skin tumors, as shown above in Figure 2B. In addition, high expression of more neutrophil marker genes including Cxcr2, Ccr1, Mmp9, and Arg2 was detected in this neutrophil cluster (Figure 6D). Overall, our data showed that neutrophils hugely expanded during skin tumor progression, highlighting their pivotal role in tumor growth. Further studies are required to fully understand the role of neutrophils during tumor progression, providing more insights on targeting neutrophils against cancer.

### 3.9. Mast Cells

Mast cells, an essential part of the immune system, act as important regulators in cancer development [66]. Various studies have revealed that mast cells can have a pro-tumorigenic or anti-tumorigenic role in different types of cancers [67]. We detected one mast cell cluster that highly expressed mast cell marker genes Gata2, Ms4a2, Mcpt4, and Fcer1a (Figure 6E). We did not observe a significant change of relative abundance in control skins and skin tumors, as shown in Figure 2B. However, we did observe activated mast cells in skin tumors shown by higher expression of Fcer1a, Il1rl1, and Tpsb2 in skin tumors (Figure 6F). Further study is required to better clarify the role of activated mast cells in relation to skin tumor progression.

### 3.10. Langerhans Cells

Langerhans cells (LCs), tissue-resident macrophages of the skin, reside in the epidermis as a dense network of immune system sentinels [68]. Their influence on the immune response is complicated, playing both pro-tumor and anti-tumor roles. Some studies showed a significant decline in the cell numbers of LCs in human melanomas and squamous cell carcinomas [69]. We identified one LC cell cluster characterized by high expression of MHCII, Langerin (Cd207), Epcam, Cd24a, and Csf1r (Figure 7A and Appendix A). Strikingly, skin tumors had a dramatic decrease in relative abundance of LCs compared to control skins (Figure 2B). In line with the absolute reduction of LCs in skin tumors, a significant reduction of Cd207 and Epcam expression was observed in skin tumors (Figure 7B). Further studies are needed to understand the mechanism causing the substantial reduction of LCs during skin tumor progression.

### 3.11. ILC2

ILC2, type 2 innate lymphoid cells, act as a crucial regulator in tumor immunity and play a promoting or suppressive role in different tumors [70,71]. We detected one ILC2 cell cluster characterized by high expression of ILC2 marker genes Gata3, Il7r, Il5, and Il13 (Figure 7C). Further, this ILC2 cell cluster dramatically decreased during skin tumor progression, as shown in Figure 2B. Consistent with this, we observed lower expression of Gata3 and Il7r in skin tumors (Figure 7D). More studies to investigate the roles of ILC2 and the mechanisms causing the reduction of ILC2 in skin tumor progression are needed.

### 3.12. DETC and γδ T Cells

γδ T cells are unconventional T cells with unique roles not restricted to MHC-mediated antigen presentation [72]. In mice, γδ T cells first develop in the embryonic thymus and further develop to distinct γδ T cells at different stages and localizations along with varying TCR pairs [73]. For instance, Vγ5^+^ dendritic epidermal T cells (DETCs) are located in the skin. However, DETCs do not exist in human skin.

We detected one DETC cell cluster shown by the expression of Trdc, Tcrg-C1, Tcrg-C2, and Tcrg-C4 (Figure 7E). We detected another γδ T cell cluster shown by the expression of Cd163l, 5830411N06Rik, and Il17a (Figure 7F). Similar to LCs and ILC2, these two populations significantly decreased in skin tumors (Figure 2B). However, the mechanisms causing this dramatic decrease of these two γδ T cell clusters remain unclear. Future studies uncovering the mechanisms will provide more insights into the function of γδ T cells in tumor immunology and their application to cancer therapy.

### 3.13. DC Subtypes in Control Skins and Skin Tumors

Dendritic cells (DCs), a diverse group of specialized antigen-presenting cells, play key roles in regulating both innate and adaptive immune responses [74]. During cancer, different DC subtypes are localized in or recruited to tumors. However, the dual and opposing roles of DC subsets as tumor-promoting and tumor-suppressing cells in cancer immunology need further investigation [75]. We detected three DC subsets, including cDC1 with a high expression of Clec9a and Xcr1, cDC2 with a high expression of Cd209a and Cd209d, and mDC with a high expression of Fscn1 and Cacnb3 (Figure 7G). We did not observe a significant change in cDC1 between control skins and skin tumors, while we observed cDC2 and mDC dramatically increased in skin tumors (Figure 2B). Further, we found a higher expression of Cxcl9, Cxcl10, and Dpp4 in DCs in skin tumors (Figure 7H). Cxcl9/Cxcl10-engineered dendritic cells were reported to promote T cell activation and enhance immunotherapy in lung cancer [76], highlighting the critical role of DCs in cancer. We will further investigate the mechanisms regulating DC subset migration into the TME during skin tumor progression and engineer a subset of DCs through Cxcl9/Cxcl10 to further improve the antitumor therapy.

## 4. Discussion

Immunotherapy with immune checkpoint inhibitors, such as cemiplimab and pembrolizumab, has revolutionized the treatment of advanced cSCC [77]. However, not all patients respond to immunotherapy treatment and some develop resistance over time [6]. Primary and acquired resistance results from complex and constant interactions between tumor cells and the surrounding tumor microenvironment. Research into targeting the TME for cSCC immunotherapy is ongoing. For example, the anti-PD1 treatment in cSCC elevates immunosuppressive Tregs, which are known to assist tumor escape in cSCC [78]. Further investigation of the TME will shed light on potential therapeutic targets and provide an opportunity to cure advanced cancers resistant to existing treatment. To this end, we applied scRNA-seq to facilitate a better understanding of the immune cell changes in the tumor microenvironment during cSCC progression. In this study, we fully characterized CD45^+^ immune cells in response to skin tumor progression in murine skin. First, we generated scRNA-seq data from CD45^+^ immune cells and identified 15 key immune cell types and revealed the overall immune responses to skin tumor progression. We further examined the molecular features of each immune cell type. These single cell profiles offer new insights on dissecting cutaneous immune response during skin tumor progression.

One key finding was a dramatic accumulation of macrophages as the most abundant cell population in skin tumors. A better understanding of molecules and pathways responsible for the recruitment of macrophages will shed light on developing macrophage-centered therapeutic strategies against cancer. Of note, our data showed a higher level of Cxcl10 and Ccl8 in the macrophage population in skin tumors. This is consistent with recent studies. One reported that tumor-intrinsic Cxc1l0 is critical for immunotherapeutic efficacy in lung, colon, and liver tumors [79]. High Cxcl10 expression is associated with better survival rates in tumor patients receiving immunotherapies. In another study, induced Ccl8 expression in macrophages is associated with colorectal cancer progression [80]. Therefore, further studies on targeting Cxcl10 and Ccl8 secreted by macrophages may provide new prospects for skin cancer therapy. Additionally, further investigation into the mechanisms by which macrophages facilitate tumor progression and immune evasion will be pursued.

Our data also identified a subset of CD4^+^ T cells that expressed cytotoxic marker genes, including Ifng, Nkg7, Lamp1, Lamp2, Klrc1, and Runx3, while lacking expression of Cd27. This suggests a potential role of this cytotoxic subset of CD4^+^ T cells in skin tumor progression. Consistently, an increasing number of studies have detected cytotoxic CD4^+^ T cells in various conditions, including acute viral infections, anti-tumor responses in human cancer patients, and chronic inflammatory responses in autoimmune diseases [54,81,82], posing important questions as to their biological and clinical relevance. More importantly, this cytotoxic subset of CD4^+^ T cells also expressed Cd70, Cd80, Cd81, and Cd83, which have been reported to be expressed in activated T cells [51,83,84]. Thus, targeting these molecules could enhance the activation and proliferation of CD4^+^ T cells, improving immune responses in cancer. Furthermore, a deeper understanding of the mechanisms and functional regulation of these cytotoxic CD4^+^ T cells will expand the immune effector toolbox and lead to more effective immunotherapies.

Furthermore, our data highlighted the importance of NK cells during skin tumor progression. Our previous work using a fluorescent tracing skin transplantation model also showed that innate immune cell deficiency permits rapid tumorigenesis [18], demonstrating the critical roles of innate immune cells during skin tumor progression. These findings are consistent with the growing evidence supporting NK cell therapy as an effective tool for cancer immunotherapy [85,86]. Since NK cell activity is controlled by many different inhibitory and activating NK receptors [87], more extensive studies on these receptors could provide valuable insights for enhancing NK cell-based immunotherapy.

In addition, our data showed a substantial reduction of LCs during skin tumor progression, highlighting the importance of LCs in tumor immunology. Consistently, this dramatic decrease of LCs was also observed in human cSCC lesions [88]. Recent studies have demonstrated the capacity of LCs to activate Cd4 and Cd8 T cell compartments, suggesting their immune-stimulatory potential. Despite the implicated role of LCs in antitumor immunity, mice with LC depletion exhibited resistance to tumor formation in a two-stage carcinogenesis model [89]. Collectively, these findings suggest that LCs play a multifaceted role in cutaneous carcinogenesis [90]. Therefore, further studies are required to better understand the role of LCs in tumor immunology, and the mechanism underlying their reduction in skin tumor progression.

While our findings provide valuable insights, our data were generated using preclinical mouse models. To address this limitation, we analyzed the publicly available human cSCC dataset GSE144236 [13] and compared our findings to it. As shown in Appendix A, we observed elevated expression of CCL8, CXCL10, MRC1, and CX3CR1 in macrophages from human cSCC patients compared to normal skins (Appendix A). Additionally, in the T cell cluster, we found increased expression of interferon pathway genes (IFNG, STAT1, IFIT2, IFIT3, ISG15, and BST2) and exhaustion marker genes (TOX, PDCD1, TIGIT, HAVCR2, LAG3, and CTLA4) in human cSCC compared to normal skins (Appendix A). These results align with our observations in preclinical mouse models. Another limitation of our study is the lack of functional assays. In future studies, we will further validate our findings in human samples and incorporate functional assays to better characterize the mechanism contributions of specific immune cell populations during skin tumor progression.

## 5. Conclusions

We utilized scRNA-seq to comprehensively characterize the immune cell landscape during skin tumor progression. Our findings reveal dramatic changes in various immune cell populations, highlighting their potential roles in promoting or restraining tumor growth. Specifically, we identified key immune cell subsets and signaling pathways that may contribute to immune evasion and tumor progression. These insights enhance our understanding of the tumor microenvironment and its immunological complexity. Ultimately, our data may provide a foundation for future studies aimed at identifying novel therapeutic targets and guiding the development of clinical applications for cancer treatment.

## Figures and Tables

**Figure 1 cancers-17-01379-f001:**
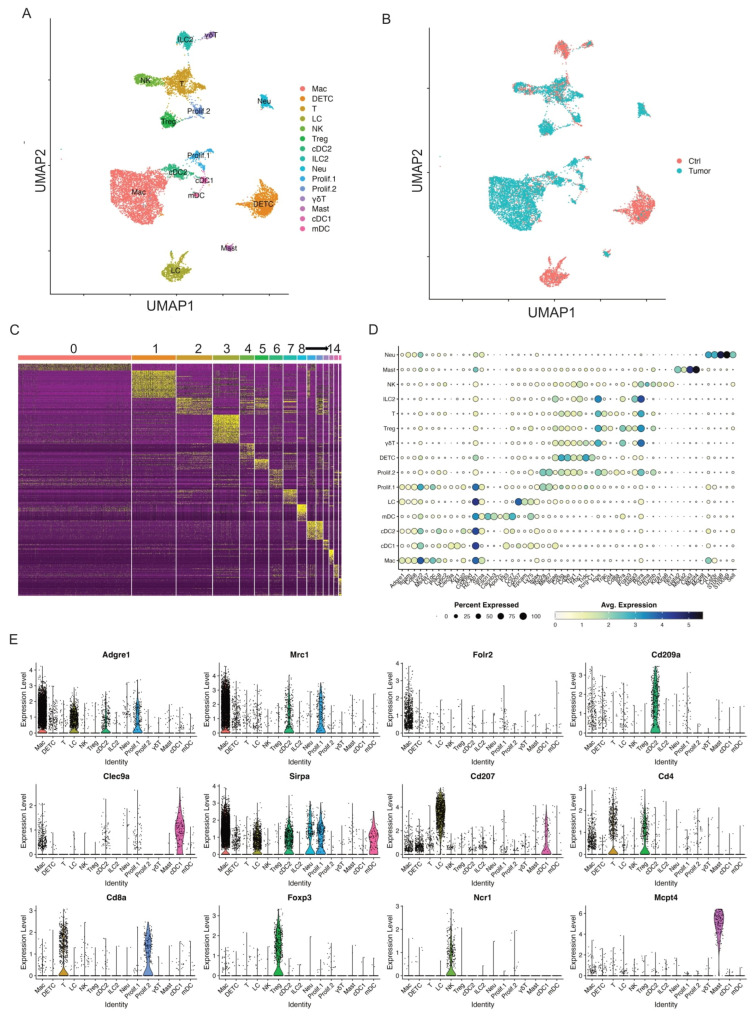
Single-cell profiling identifies key immune cell clusters in murine skin. (**A**) UMAP visualization of 11,525 skin CD45^+^ immune cells identified 15 clusters after unsupervised clustering. (**B**) UMAP projection of all skin immune cell populations colored by experimental conditions (red: Ctrl and blue: Tumor). (**C**) Heat map showing the top 20 most differentially expressed genes in each cluster identified by unsupervised clustering of skin immune cells. Cluster 0: Mac; Cluster 1: DETC; Cluster 2: T cells; Cluster 3: LC; Cluster 4: NK; Cluster 5: Treg; Cluster 6: cDC2; Cluster 7: ILC2; Cluster 8: Neu; Cluster 9: Prolif.1; Cluster 10: Prolif.2; Cluster 11: γδT; Cluster 12: Mast; Cluster 13: cDC1; Cluster 14: mDC. (**D**) Dot plot for immune cell population marker genes in different clusters. Marker transcript expression levels (*x* axis) for the 15 immune cell populations (*y* axis). Size of dots represents the fraction of cells expressing a particular marker, and color intensity indicates mean normalized scaled expression levels. (**E**) Violin plots show normalized expression distribution on a per cluster basis for selected immune cell population marker genes that distinguish major populations: Adgre1, Mrc1, and Folr2 for macrophage; Cd209a for cDC2; Clec9a for cDC1; Sirpa in macrophages, mDC, and cDC2; Cd207 for LC; Cd4, and Cd8a for T cells; Foxp3 for Tregs; Ncr1 for NK cells; Mcpt4 for mast cells.

**Figure 2 cancers-17-01379-f002:**
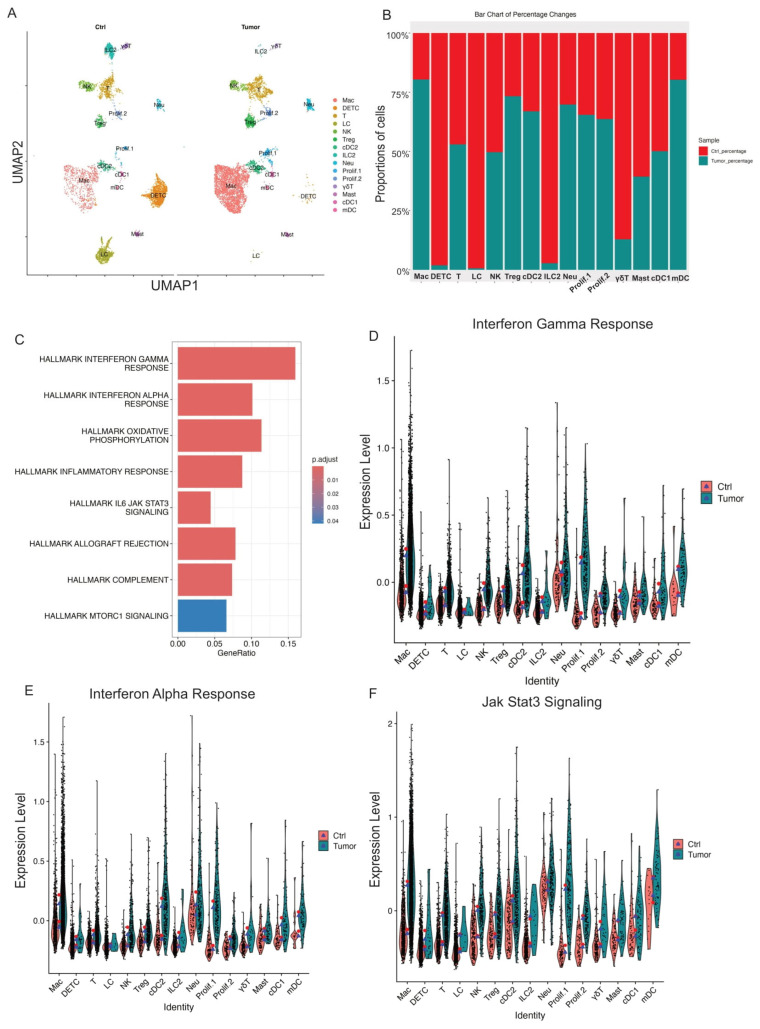
Skin tumor progression induces APC-dominant shifts and reprograms CD45^+^ cell transcriptomes toward interferon response and JAK/STAT3 signaling. (**A**) UMAP projection of all immune cell clusters across different conditions (Ctrl: control skins; Tumor: skin tumors). (**B**) Relative proportions for each immune cell population for control skins and skin tumors. The immune cell percentage in control skins is shown in red, whereas in skin tumors, it is shown in blue. (**C**) GSEA querying hallmark genes depicting significant enrichment of Interferon response and JAK/STAT3 signaling in skin tumors (HALLMARK_INTERFERON_GAMMA_RESPONSE, HALLMARK_INTERFERON_ALPHA_RESPONSE, HALLMARK_IL6_JAK_STAT3_SIGNALING). (**D**) Expression level of interferon gamma response genes in Ctrl and Tumor showing higher interferon gamma response in skin tumors. The red dot shows mean expression in Ctrl, while blue triangles show median expression in Tumor. (**E**) Expression level of interferon alpha response genes in Ctrl and Tumor showing higher interferon alpha response in skin tumors. (**F**) Expression level of Jak/STAT3 genes in Ctrl and Tumor showing increased Jak/STAT3 signaling in skin tumors.

**Figure 3 cancers-17-01379-f003:**
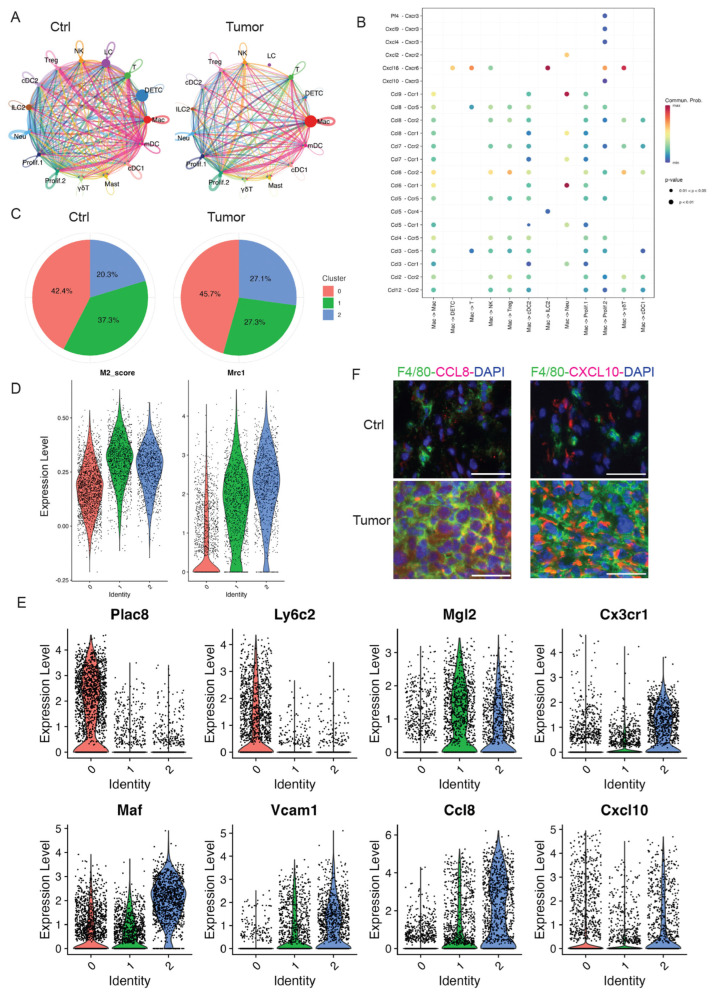
Macrophage dramatically increased in skin tumors. (**A**) The circle diagram illustrates the signal crosstalk of all cells in our dataset, with the thickness representing the signal strength. Each color represents a different cell type, with macrophages shown in red. (**B**) The significant ligand–receptor pairs that contribute to the signaling transmission from macrophages to other cell types. Dots represent the contribution of each receptor pair in signals emitted by macrophages towards various cells in our dataset. Dot size indicates significance, while color shade represents the magnitude of contribution. Darker shades, particularly red, indicate a higher contribution. UMAP projection of three sub-clusters of macrophages. (**C**) The pie charts of percentage of three subclusters in Ctrl and Tumor showing sub-cluster 2 dramatically increased in Tumor. (**D**) Higher expression of M2 gene score and Mrc1 in sub-cluster 1 and 2 suggestive of M2-like macrophages. (**E**) Violin plots showing gene expression in three sub-clusters. Plac8 and Ly6c2 are highly expressed in sub-cluster 0. Expression level of M2 macrophage marker genes including Mgl2, Cx3cr1, Vcam1, Maf, Ccl8, and Cxcl10 in three sub-clusters of macrophages. (**F**) Immunofluorescent staining in control skins (Ctrl) and 2323 skin tumors (Tumor) showing a higher expression of Ccl8 and Cxcl10 in the macrophages in skin tumors compared to control skins. Scales: 100 px.

**Figure 4 cancers-17-01379-f004:**
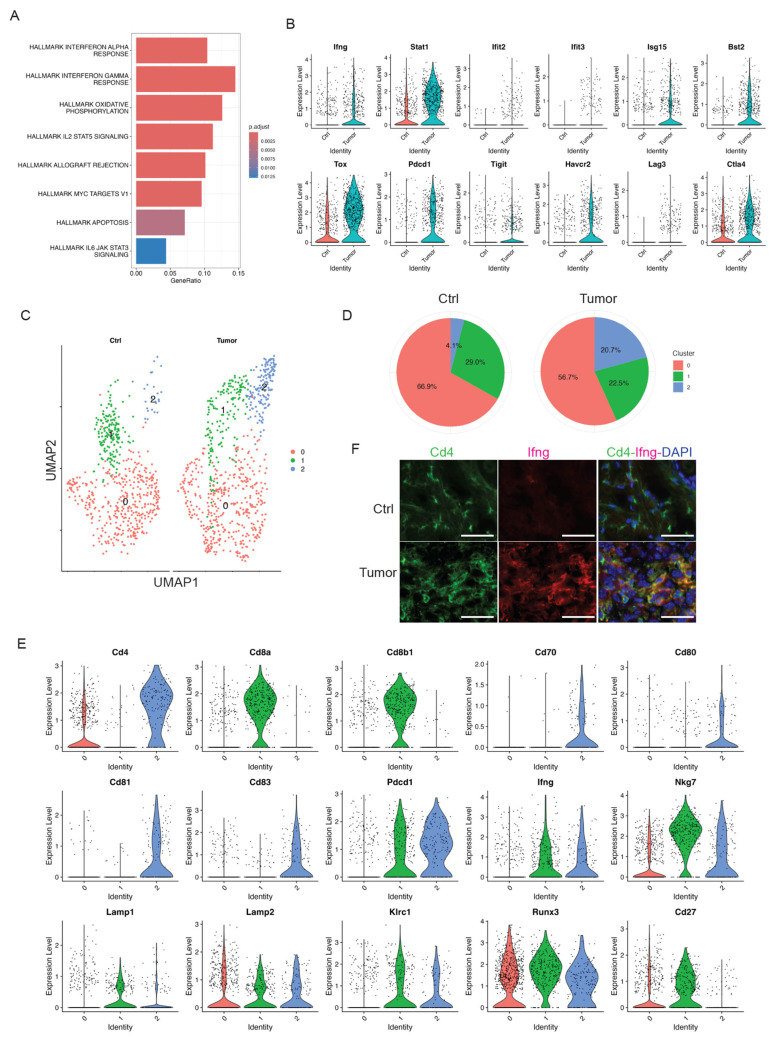
Skin tumor progression induces CD4^+^ activated T cells. (**A**) GSEA querying hallmark genes depicting significant enrichment of Interferon response in skin tumors (HALLMARK_INTERFERON_GAMMA_RESPONSE and HALLMARK_INTERFERON_ALPHA_RESPONSE). (**B**) Expression level of interferon pathway genes including Ifng, Stat1, Ifit2, Ifit3, Isg15, and Bst2, and exhaustion marker genes including Tox, Pdcd1, Tigit, Havcr2, Lag3, and Ctla4 in Ctrl and Tumor. (**C**) UMAP projection of three sub-clusters of the T cell cluster: three overlapped sub-clusters on the left and three sub-clusters in Ctrl and Tumor on the right. (**D**) Percentages of sub-clusters of the T cell cluster in control skins and skin tumors showing a dramatic increase in sub-cluster 2 in skin tumors. (**E**) Expression level of Cd4, Cd8a, Cd8b1, Cd70, Cd80, Cd81, Cd83, Pdcd1, Ifng, Nkg7, Lamp1, Lamp2, Klrc1, Runx3, and Cd27 in three sub-clusters of the T cell cluster showing sub-cluster 2 is a CD4^+^ T cell cluster. A higher level of co-stimulatory molecules including Cd70, Cd80, Cd81, and Cd83 was detected in sub-cluster 2. Additionally, co-inhibitory molecule Pdcd1 was highly expressed in sub-cluster 2, consistent with activated T cells in skin tumors. The cytotoxic CD4^+^ T cell marker genes such as Ifng, Nkg7, Lamp1, Lamp2, Klrc1, and Runx3 were highly expressed in subcluster 1 and 2. Downregulation of Cd27 was observed in sub-cluster 2. (**F**) Immunofluorescent staining in control skins (Ctrl) and 2323 skin tumors (Tumor) showing a co-localization of Cd4 and Ifng in skin tumors but not in control skins. Scale bars: 100 px.

**Figure 5 cancers-17-01379-f005:**
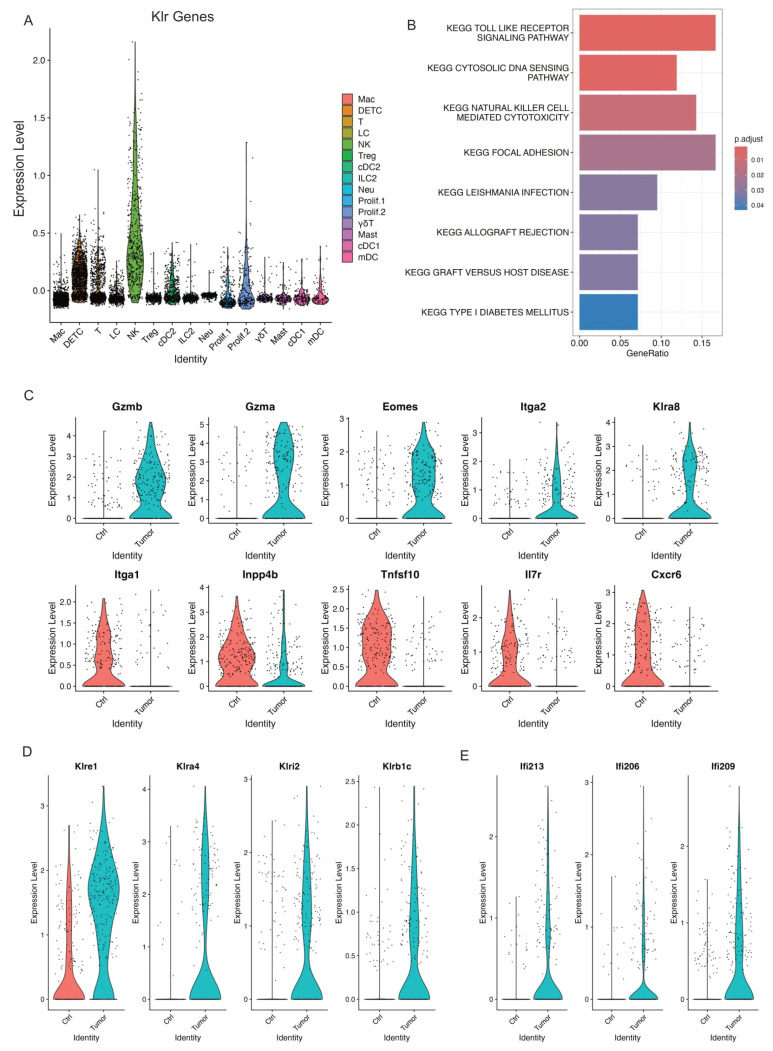
Activated NK cells in skin tumors. (**A**) Expression level of Klr genes across all the immune cell clusters validating the NK cell cluster. (**B**) GSEA querying hallmark genes depicting significant enrichment of NK cell mediated cytotoxicity in skin tumors (KEGG_NATURRAL_KILLER_CELL_MEDIATED_CYTOTOXICITY). (**C**) Violin plots showing higher expression of activated NK genes including Gzmb, Gzma, Eomes, Itga2, and Klra8 in skin tumors indicated activated NK cells in skin tumors. Consistently, a higher expression of NK inactivated genes such as Itga1, Inpp4b, Tnfsf10, Il7r, and Cxcr6 in control skins. (**D**) Violin plots showing specific NK receptors including Klre1, Klra4, Klri2, and Klrb1c highly expressed in skin tumors. (**E**) Violin plots showing higher expression of interferon genes including Ifi213, Ifi206, and Ifi209 in skin tumors.

**Figure 6 cancers-17-01379-f006:**
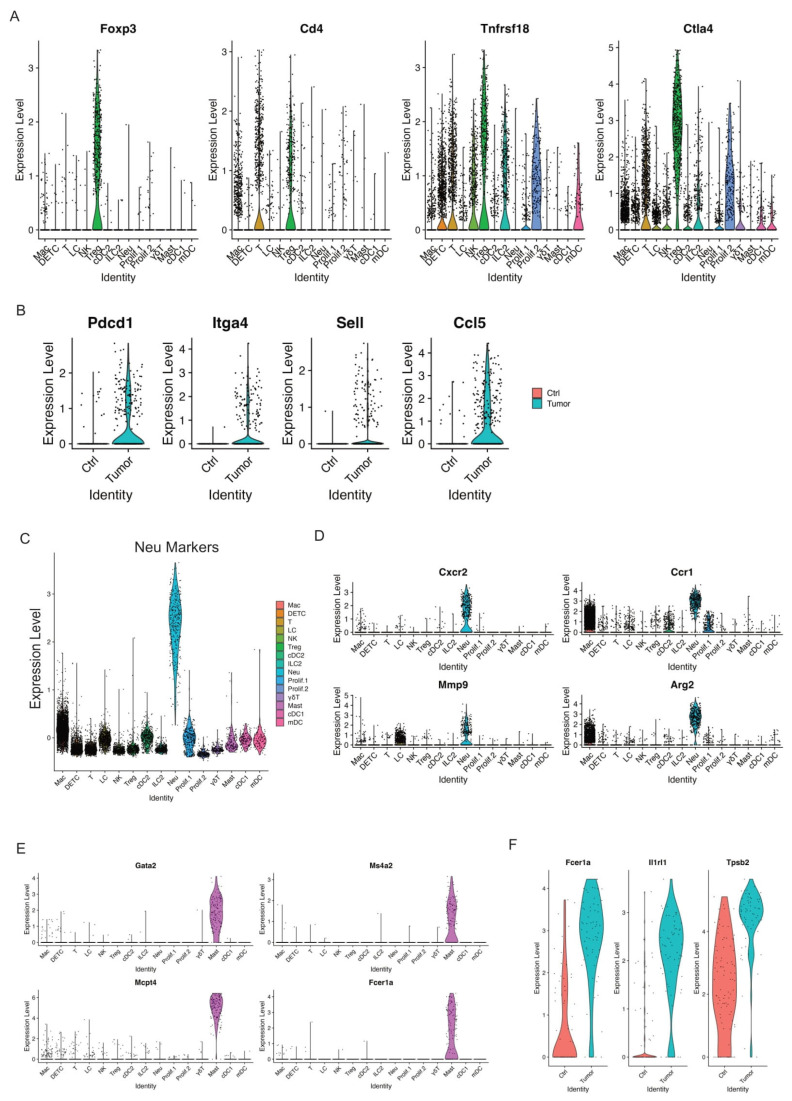
Increased Tregs and activated mast cells in skin tumors. (**A**) Violin plots showing the expression of Treg marker genes Foxp3, CD4, Tnfrsf18, and Ctla4 across immune cell clusters. (**B**) Expression of Pdcd1, Itga4, Sell, and Ccl5 was highly expressed in skin tumors. (**C**) Expression level of Neu marker genes across immune cell clusters validating the Neu cell cluster. (**D**) Expression level of Cxcr2, Ccr1, Mmp9, and Arg2 across immune cell clusters showing a higher expression of these genes in the Neu cell cluster. (**E**) Mast cell marker genes such as Gata2, Ms4a2, Mcpt4, and Fcer1a selectively expressed in Mast cell cluster. (**F**) Expression level of Fcer1a, Ir1rl1, and Tpsb2, the activation genes of mast cells, in control skins and skin tumors showing activated mast cells in skin tumors.

**Figure 7 cancers-17-01379-f007:**
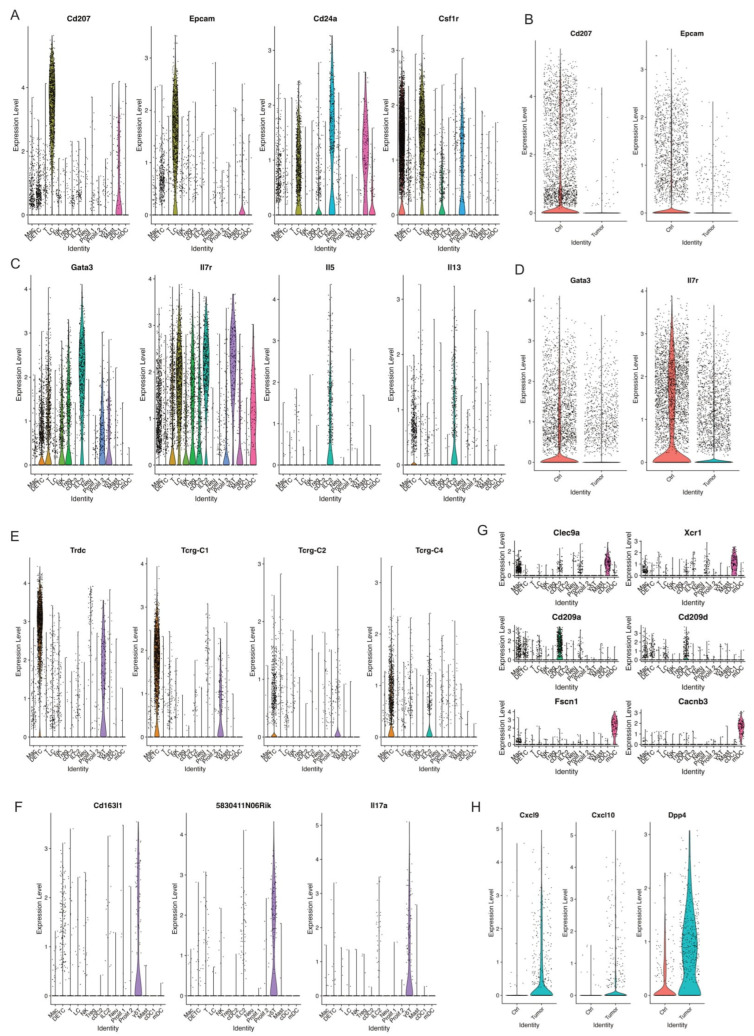
Decreased populations of DETC, γδT, and ILC2 in skin tumors. (**A**) Violin plots showing the expression of LC marker genes Cd207, Epcam, Cd24a, and Csf1r across immune cell clusters. (**B**) Violin plots showing higher level of Cd207 and Epcam in Ctrl. (**C**) Expression level of ILC2 marker genes Gata3, Il7r, Il5, and Il13 across immune cell clusters defined ILC2 cell cluster. (**D**) A higher expression of ILC2 marker genes Gata3 and Il7r was detected in Ctrl. (**E**) Expression level of Trdc, Tcrg-C1, Tcrg-C2, and Tcrg-C4 in DETC and γδT. (**F**) Expression level of γδT marker genes Cd163l1, 5830411N06Rik, and Il7a indicating γδT cells. (**G**) Expression level of cDC1 marker genes Clec9a and Xcr1, cDC2 marker genes Cd209a and Cd209d, mDC marker genes Fscn1 and Cacnb3 across immune cell clusters. (**H**) Expression level of Cxcl9, Cxcl10, Dpp4 in Ctrl and Tumor.

## Data Availability

The single-cell RNA sequencing data from mice analyzed in this study are publicly available in the Gene Expression Omnibus (GEO) under accession number GSE280070. For the human cSCC dataset, this study analyzed publicly available scRNA-seq data accessible through GEO under accession number GSE144236.

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
