# Peer review of "Single-Cell Profiling Reveals Global Immune Responses During the Progression of Murine Epidermal Neoplasms"

_cancers, 2025, doi:10.3390/cancers17081379_

Round 1
Reviewer 1 Report
Comments and Suggestions for Authors
The study conducted an analysis of the role played by immune cells within the tumor microenvironment during the progression of skin tumors. Utilizing single-cell level detection, the study identified 13 distinct immune cell types in both the skin and tumor microenvironments of mice. Additionally, the study discerned 15 CD45+ immune cell clusters, which included macrophages, Langerhans cells (LC), conventional type 1 dendritic cells (cDC1), conventional type 2 dendritic cells (cDC2), migratory/mature dendritic cells (mDC), epidermal dendritic T cells (DETC), dermal gamma delta T cells (γδT), T cells, regulatory T cells (Tregs), natural killer cells (NK), type 2 innate lymphoid cells (ILC2), neutrophils (Neu), mast cells (Mast), and two proliferating populations (Prolif.1 and Prolif.2). As skin tumors advanced, there was a notable increase in the relative proportion of macrophages, cDC2, mDC, Tregs, and Neu, with macrophages emerging as the predominant immune cell population within the tumor. These macrophages served as a primary communication "hub" within the tumor microenvironment. Conversely, the percentages of DETC, gamma delta T, ILC2, and LC diminished during tumor progression. Furthermore, the advancement of skin tumors led to a significant upregulation of Jak2/Stat3 expression and an interferon response. This was accompanied by the activation of T cells and NK cells, as evidenced by elevated expression levels of IFN-γ and Granzyme B. Notably, the pronounced infiltration of M2-like macrophages and Tregs created an immunosuppressive microenvironment, aligning with the heightened expression of the Stat3 pathway observed in skin tumors. In conclusion, this study delineates the composition of immune cells within skin tumors, offering a nuanced understanding of the immune response during their progression. Furthermore, it unveils novel insights into the mechanisms of cancer immune evasion. This manuscript demonstrates a certain level of innovation, and its subject matter aligns with the journal's requirements. It is recommended for acceptance following revision.
- Tumor microenvironment and heterogeneity are critical factors influencing the efficacy of immunotherapy for skin tumors. The following reference focuses on the tumor microenvironment and heterogeneity in skin cancer, providing detailed insights into single-cell analysis and multi-omics approaches. These aspects should be cited by the author in the background section to support the relevance of these methodologies (A. Gu, J. Li, M.-Y. Li, Y. Liu, Patient-derived xenograft model in cancer: establishment and applications. MedComm, 2025, 6, e70059. DOI: 10.1002/mco2.70059).
- Further research is warranted to elucidate the mechanisms by which macrophages facilitate tumor progression and immune evasion, particularly through their interplay with tumor cells. This may include a detailed analysis of their metabolic processes, signaling pathways, and collaborations with other immune entities within the tumor microenvironment.
It would be beneficial to conduct a more in-depth statistical analysis of the immune cell types that exhibited less pronounced changes in ratio between skin tumor and control skin. This will help ascertain the statistical significance of these alterations, thereby reinforcing the validity of the results.
- The study could incorporate a comprehensive discussion regarding the implications of current immunotherapeutic strategies within the context of the skin tumor microenvironment. This would include an analysis of potential challenges that may arise and the development of proposed solutions accordingly. By providing this guidance, it would significantly contribute to future research endeavors and clinical applications.
- Future research endeavors are encouraged to delve into the roles of noncoding RNAs and epigenetic modifications within the tumor microenvironment, including but not limited to the regulation of microRNAs and alterations in DNA methylation patterns. Such investigations will promote a comprehensive comprehension of the immune escape mechanisms inherent to skin tumors.
- The author needs to include a lot of literature in the introduction, and it is recommended to cite the following literature:
[1] M. Yang, J. Zhou, L. Lu, D. Deng, J. Huang, Z. Tang, X. Shi, P.-C. Lo, J. F. Lovell, Y. Zheng, H. Jin, Tumor cell membrane-based vaccines: A potential boost for cancer immunotherapy. Exploration 2024, 4, 20230171. https://doi.org/10.1002/EXP.20230171
[2] Ji L., Zhang H., Tian G., et al., (2023). Tumor microenvironment interplay amid microbial community, host gene expression and pathological features elucidates cancer heterogeneity and prognosis risk. The Innovation Life 1(2), 100028. https://doi.org/10.59717/j.xinn-life.2023.1
[3] Z. Zhang, D. Huang, J. Feng, W. Li, Z. Wang, M. Lu, Y. Luo, W. Yang, Z. Xu, Q. Xie, W. Ding, X. Tan, W. He, G. Li, H. Liu, S. Lei, PDE5 inhibitors against cancer via mediating immune cells in tumor microenvironment: AI-based approach for future drug repurposing exploration. Interdiscip. Med. 2024, 2, e20230062. https://doi.org/10.1002/INMD.20230062
Author Response
1. Tumor microenvironment and heterogeneity are critical factors influencing the efficacy of immunotherapy for skin tumors. The following reference focuses on the tumor microenvironment and heterogeneity in skin cancer, providing detailed insights into single-cell analysis and multi-omics approaches. These aspects should be cited by the author in the background section to support the relevance of these methodologies (A. Gu, J. Li, M.-Y. Li, Y. Liu, Patient-derived xenograft model in cancer: establishment and applications. MedComm, 2025, 6, e70059. DOI: 10.1002/mco2.70059)
Response: Thanks for pointing this out. We have included this reference in line 64.
2. Further research is warranted to elucidate the mechanisms by which macrophages facilitate tumor progression and immune evasion, particularly through their interplay with tumor cells. This may include a detailed analysis of their metabolic processes, signaling pathways, and collaborations with other immune entities within the tumor microenvironment.
Response: We appreciate the reviewer’s suggestion on elucidating the mechanisms by which macrophages facilitate tumor progression and immune evasion. In the future studies, we will further investigate the mechanisms through a detailed analysis of their metabolic processes, signaling pathways, and collaborations with other immune entities within the tumor microenvironment. We have included this in lines 631 to 633.
It would be beneficial to conduct a more in-depth statistical analysis of the immune cell types that exhibited less pronounced changes in ratio between skin tumor and control skin. This will help ascertain the statistical significance of these alterations, thereby reinforcing the validity of the results.
Response: We appreciate the reviewer’s suggestion to conduct a more in-depth statistical analysis of immune cell types with less pronounced changes in ratio between skin tumor and control skin. To clarify, we performed differential abundance analysis using the default statistical methods implemented in Seurat, which include the Wilcoxon rank-sum test for differential expression and log-normalization for data scaling. To ensure robustness, we applied default multiple testing corrections using the Benjamini-Hochberg false discovery rate (FDR) adjustment to control for false positives. Differences in immune cell populations between control and tumor tumors were assessed using chi-square test. These statistical approaches provide a reliable framework for detecting differences in immune cell populations, even when the changes in proportion are subtle. To clarify these details, we have updated the manuscript to describe the statistical methods used in our analysis in lines 177-185.
3. The study could incorporate a comprehensive discussion regarding the implications of current immunotherapeutic strategies within the context of the skin tumor microenvironment. This would include an analysis of potential challenges that may arise and the development of proposed solutions accordingly. By providing this guidance, it would significantly contribute to future research endeavors and clinical applications.
Response: We appreciate the reviewer’s suggestion to discuss the implications of current immunotherapeutic strategies within the context of the skin tumor microenvironment. In the revised manuscript, we have expanded our discussion to highlight the potential implications of future research endeavors and clinical applications. This discussion can be found in lines 605-613.
4. Future research endeavors are encouraged to delve into the roles of noncoding RNAs and epigenetic modifications within the tumor microenvironment, including but not limited to the regulation of microRNAs and alterations in DNA methylation patterns. Such investigations will promote a comprehensive comprehension of the immune escape mechanisms inherent to skin tumors.
Response: Thank you for this great suggestion. We totally agree the importance of the epigenetic modifications within the tumor microenvironment. Since scRNA-seq is primarily designed to analyze gene expression level, it is not directly suited for microRNA and DNA methylation analysis. We will use other appropriate tools to investigate the epigenetic modifications in skin tumors in the future.
5. The author needs to include a lot of literature in the introduction, and it is recommended to cite the following literature:
[1] M. Yang, J. Zhou, L. Lu, D. Deng, J. Huang, Z. Tang, X. Shi, P.-C. Lo, J. F. Lovell, Y. Zheng, H. Jin, Tumor cell membrane-based vaccines: A potential boost for cancer immunotherapy. Exploration 2024, 4, 20230171. https://doi.org/10.1002/EXP.20230171
[2] Ji L., Zhang H., Tian G., et al., (2023). Tumor microenvironment interplay amid microbial community, host gene expression and pathological features elucidates cancer heterogeneity and prognosis risk. The Innovation Life 1(2), 100028. https://doi.org/10.59717/j.xinn-life.2023.1
[3] Z. Zhang, D. Huang, J. Feng, W. Li, Z. Wang, M. Lu, Y. Luo, W. Yang, Z. Xu, Q. Xie, W. Ding, X. Tan, W. He, G. Li, H. Liu, S. Lei, PDE5 inhibitors against cancer via mediating immune cells in tumor microenvironment: AI-based approach for future drug repurposing exploration. Interdiscip. Med. 2024, 2, e20230062. https://doi.org/10.1002/INMD.20230062
Response: We have added the suggested references in lines 46 and 58, except for the reference titled ‘Tumor cell membrane-based vaccines: A potential boost for cancer immunotherapy’. Our study focuses specifically on immune cell populations, whereas this reference primarily discusses tumor cell membrane-based vaccines, which fall outside the scope of our analysis.
Reviewer 2 Report
Comments and Suggestions for Authors
This study investigates the immune profile during the progression of murine cSCC using scRNA-seq. The authors identify 15 distinct immune cell clusters and report significant shifts in the tumor microenvironment, including an increase in macrophages, Tregs, and neutrophils and a decrease in Langerhans cells, γδ T cells, and ILC2. Their data shows that M2 macrophages are predominant communication hubs within the tumor, and the Jak2/Stat3 and interferon response pathways are upregulated. The findings suggest that macrophage-mediated immunosuppression and T cell exhaustion contribute to tumor progression, offering insights into potential targets for immunotherapy.
While the study provides comprehensive profiling and valuable insights into the immune microenvironment of skin tumors, it is limited by its exclusive use of a murine model without validation in human samples. Furthermore, mechanistic conclusions, particularly those concerning the role of macrophages and cytotoxic CD4+ T cell, are based solely on transcriptomic data and lack functional validation.
Major points:
- The authors should acknowledge the absence of human cSCC data as a limitation and consider including validation studies in human samples.
- Functional assays would strengthen the mechanistic claims. These should be discussed as limitations or directions for future work.
Minor points:
- Please include scale bars for all immunofluorescence (IF)-stained images.
- In Figure 3F, the authors should add IF-stained images of normal skin to compare Ccl8 and Cxcl10 expression in macrophages.
- In line 407, the authors should clarify the statement: “We identified a distinct T-cell cluster expressing Cd3d/e/g, Cd4, Cd8a, and Thy1 (Figure 1D). We did not observe a significant infiltration of this T cell cluster in skin tumors as shown above in Figure 2B.” It is unclear which T cell subtype this refers to.
- In Figure 4B, it should be clarified whether the gene expression levels (e.g., IFN-γ, exhaustion markers) are specific to T cells or represent immune cell data.
- In Figure 4F, please include IF-stained images of normal skin to enable comparison of IFN-γ expression by CD4+ T cells.
- In line 618, the authors refer to “various conditions” in which cytotoxic CD4+ T cells have been detected. Please specify what these conditions are.
- The importance of Langerhans cells in tumor immunology should be discussed, particularly in the context of their observed reduction during tumor progression.
Author Response
Major points:
- The authors should acknowledge the absence of human cSCC data as a limitation and consider including validation studies in human samples.
Response: Thank you for this insightful comment. We acknowledge the absence of human cSCC data as a limitation of our study. To address this, we compared our findings with data from human cSCC patients using a publicly available dataset GSE144236. As shown in the newly added Supplemental Figure 5, we observed a higher expression of CCL8, CXCL10, MRC1, and CX3CR1 in the macrophages of human cSCC patients compared to normal skins. Also, we observed a higher expression of interferon pathway genes including IFNG, STAT1, IFIT2, IFIT3, ISG15, and BST2, and exhaustion marker genes including TOX, PDCD1, TIGIT, HAVCR2, LAG3, and CTLA4 in the T cell sub-cluster in human cSCC compared to normal skins. These results are consistent with our findings in preclinical mouse models. But this human dataset was generated from patient tumors without specific enrichment for immune cells and some immune cell populations are underrepresented due to limitations in cell numbers. So, we cannot compare all our data to this human dataset. We will further validate our findings in human samples in the future study. We have included these points in lines 665-674.
- Functional assays would strengthen the mechanistic claims. These should be discussed as limitations or directions for future work.
Response: We appreciate the reviewer’s suggestion regarding functional assays. We totally agree that function assays would strengthen the mechanistic claims. In lines 674-677 of the revised manuscript, we have included a discussion highlighting functional assays as our limitations and future studies.
Minor points:
- Please include scale bars for all immunofluorescence (IF)-stained images.
Response: Thank you for pointing this out. We have added the scale bars for all IF images. Please see the revised Figures 3F and 4F.
- In Figure 3F, the authors should add IF-stained images of normal skin to compare Ccl8 and Cxcl10 expression in macrophages.
Response: Thank you for the suggestion. We have added the IF images of normal skin in the revised Figure 3F.
- In line 407, the authors should clarify the statement: “We identified a distinct T-cell cluster expressing Cd3d/e/g, Cd4, Cd8a, and Thy1 (Figure 1D). We did not observe a significant infiltration of this T cell cluster in skin tumors as shown above in Figure 2B.” It is unclear which T cell subtype this refers to.
Response: Here, the T cell cluster refers to the pan T cell cluster, which expressed marker genes Cd3d/e/g, Cd4, Cd8a, and Thy1. In Figures 4C, 4D, and 4E, we showed that sub-cluster 2, characterized by a high expression of Cd4, dramatically increased in skin tumors. To improve clarity, we have updated the text to state “pan T cell cluster” in line 415.
- In Figure 4B, it should be clarified whether the gene expression levels (e.g., IFN-γ, exhaustion markers) are specific to T cells or represent immune cell data.
Response: Thank you for pointing this out. To clarify, we have specified that the gene expression levels (e.g. IFN-γ, exhaustion markers) are specific to T cells by adding “in the T cell cluster” in lines 418 and 420.
- In Figure 4F, please include IF-stained images of normal skin to enable comparison of IFN-γ expression by CD4+T cells.
Response: Thank you for the suggestion. We have added the IF images of normal skin in the revised Figure 4F.
- In line 618, the authors refer to “various conditions” in which cytotoxic CD4+T cells have been detected. Please specify what these conditions are.
Response: We have specified these conditions in the revised manuscript. In lines 638-639, we have included “in various conditions, including acute viral infections, anti-tumor responses in human cancer patients, and chronic inflammatory responses in autoimmune diseases”.
- The importance of Langerhans cells in tumor immunology should be discussed, particularly in the context of their observed reduction during tumor progression.
Response: We appreciate the reviewer’s suggestion to discuss the role of Langerhans cells in tumor immunology. In the revised manuscript, we have expanded our discussion to highlight the role of LCs during skin tumor progression. This discussion can be found in lines 655-664.